# The Impact of Oxidative Stress on Pediatrics Syndromes

**DOI:** 10.3390/antiox11101983

**Published:** 2022-10-05

**Authors:** Ginevra Micangeli, Michela Menghi, Giovanni Profeta, Francesca Tarani, Alessandro Mariani, Carla Petrella, Christian Barbato, Giampiero Ferraguti, Mauro Ceccanti, Luigi Tarani, Marco Fiore

**Affiliations:** 1Department of Maternal Infantile and Urological Sciences, Sapienza University of Rome, 00185 Rome, Italy; 2Department of Internal, Anesthesiologic and Cardiovascular Clinical Sciences, Sapienza University of Rome, 00185 Rome, Italy; 3Institute of Biochemistry and Cell Biology, IBBC—CNR, 000185 Rome, Italy; 4Department of Experimental Medicine, Sapienza University of Rome, 00185 Rome, Italy; 5SITAC, Società Italiana per il Trattamento dell’Alcolismo, 00184 Rome, Italy; 6SIFASD, Società Italiana Sindrome Feto-Alcolica, 00184 Rome, Italy

**Keywords:** oxidative stress, antioxidant, pediatrics, FASD, genetic syndrome

## Abstract

Oxidative stress is a condition determined by an imbalance between antioxidant and oxidative factors. Oxidative stress can have serious consequences on our organism. Indeed, it causes both necrosis and cell apoptosis, determining cellular aging, increased carcinogenesis, vascular stiffening, increased autoimmune diseases, and muscle decay. In the context of pediatric syndromes, oxidative stress could play a role in the first order. In fact, our review of the literature showed that in some pathologies, such as fetal alcohol spectrum disorders, oxidative stress related to the intake of ethanol during pregnancy is a main etiological factor determining the associated clinical syndrome. On the contrary, in Williams syndrome, Down syndrome, Marfan syndrome, Gaucher syndrome, ataxia-telangiectasia, autistic spectrum disorder, Fanconi’s anemia, and primitive immunodeficiencies, the increase in oxidative stress is directly associated with the genetic alterations that cause the same pathologies. Although further studies are needed to better understand the relationship between oxidative stress and pediatric diseases, a better knowledge of this crucial issue encourages future therapeutic strategies.

## 1. Introduction

Oxidative stress is a condition in which the balance between antioxidant and proxying factors is altered in favor of the production of oxidizing species; this determines the overproduction of radical oxygen species, defined as ROS, and can be associated with a reduction in the synthesis of antioxidant factors, as shown in Figure 1 [1,2]. In the correct balance of the cellular oxidative state, ROS are necessary for the performance of some functions [3]. In fact, among the functions performed by ROS, there are the degradation of pathogens, the regulation of cardiac and vascular activities, the regulation of intracellular calcium concentration, and the phosphorylation or the dephosphorylation of proteins [3,4]. In the context of pediatric syndromes, the role of oxidative stress is not yet fully elucidated, and there is no review of literature that compares the various types of oxidative stress in different syndromes [5].

For this reason, we have examined pediatric syndromes that are related to an alteration of oxidative stress, such as fetal alcohol spectrum disorders (FASD), Williams syndrome, Down syndrome, Marfan syndrome, Gaucher syndrome, ataxia–telangiectasia, autistic spectrum disorders, Fanconi’s anemia, and primitive immunodeficiencies [5,6,7,8,9].

### 1.1. Oxidative Stress and Reactive Oxygen Species

Reactive oxygen species are physiologically produced by cellular metabolism and can be divided into the following two categories: free radicals and non-radicals [10]. The molecules containing one or more unpaired electrons are defined as free radicals with high cellular reactivity, while molecules that share their own unpaired electrons are defined as non-radical species [11,12]. The three most important types of ROS are the superoxide anion (O_2_^•−^), the hydroxyl radical (^•^OH), and hydrogen peroxide (H_2_O_2_) as shown in Table 1 [13].

Hydrogen peroxide is formed thanks to three different mechanisms, such as reduced nicotinamide adenine dinucleotide phosphate (NADPH) oxidase, xanthine oxidase, or dismutation of the anion by superoxide dismutase (SOD) [14,15]. The superoxide anion O_2_^•−^, a radical species with relative stability, is formed by the transfer of an electron to the O_2_ and is therefore considered to be the primary ROS from which all the other radical species would then be formed [3]. This reaction can be mediated by the following: NADPH oxidase, xanthine oxidase, or the mitochondrial electron transport chain [12,16]. Superoxide is mainly produced at the mitochondrial level. Usually, electrons enter the mitochondrial electron transport chain to reduce oxygen in water, but 1% to 3% of the electrons are lost, forming the superoxide anion [14,17]. The NADPH oxidase is also present in neutrophilic granulocytes, monocytes, and macrophages and serves to produce the superoxide anion, which has bactericidal activity [18]. The hydroxyl radical ^•^OH is the most reactive species and can damage proteins, lipids, and carbohydrates, and cause breaks in the DNA double helix [19].

The hydroxyl radical is produced starting from H_2_O_2_ in the Haber–Weiss reaction, which requires the presence of metals such as iron or copper [23].

In the first part, there is the reduction of ferric ions to ferrous ions as follows:Fe^3+^ + O_2_^•^^−^⇒ Fe^2+^ + O_2_
while in the second phase of the reaction, known as the Fenton reaction, there is the formation of the reactive species·OH as follows:Fe^2+^ + H_2_O_2_ ⇒ Fe^3 +^ + OH^−^ + OH

The Haber–Weiss reaction can also be triggered during the inflammation process after ferritin releases iron [29,30,31].

Mitochondria favor the production of ROS in the following two conditions: when the production of ATP exceeds the energy requirements of the cell and this determines a reduction in the activity of the electron transport chain; or when the decoupling of some complexes of the electron transport chain occurs, which occurs under stressful conditions [19,32].

The superoxide anion is produced by complex I (NADH dehydrogenase) and by complex III (ubiquinone cytochrome c reductase) of the mitochondrial electron transport chain. In fact, the transfer of electrons from complex I or II to ubiquinone (Q) causes the formation of reduced ubiquinone QH_2_ [25,33,34]. QH_2_ reforms ubiquinone through a particularly reactive semiquinone anion (Q-) that transfers electrons to oxygen with the formation of a superoxide anion [25,35]. In peroxisomes, on the other hand, the transfer of electrons to oxygen causes the formation of hydrogen peroxide [36]. The same granulocytes contribute to the formation of ROS. In fact the neutrophils during the inflammatory response degrade the hydrogen peroxide through the myeloperoxidase (MPO); this enzyme, as well as the eosinophilic peroxidase, can modify the proteins through halogenation, and nitration by oxidizing tyrosine to tyrosyl radical [37,38,39].

The peroxyl radical (ROO) is another species belonging to the ROS that plays an important role in lipid peroxidation. In fact, the hydroperoxide radical (HOO) transforms polyunsaturated fatty acids into hydroperoxide (HO_2_) [20,40]. This radical is particularly reactive and promotes the formation of secondary products such as aldehyde and malondialdehyde, which are particularly harmful [40,41].

In addition to endogenous production, ROS are also formed due to external factors and unhealthy lifestyles. A major exogenous source of ROS production is cigarette smoke. Indeed, this through combustion allows the formation of compounds such as super-red anion and nitroxide [42,43]. In addition, smoke inhalation favors the recall of inflammatory cells such as neutrophils and macrophages, which increases the production of ROS [44,45].

Moreover, exposure to ozone compounds causes lipid peroxidation and induces the migration of neutrophils into the epithelium of the airways. Even short-term exposure to ozone can cause an increase in systemic inflammation with an increase in the synthesis of MPO, eosinophilic cationic proteins, and lactic dehydrogenase, resulting in alterations in lung function [38,40].

Ionizing radiation is a further source of free radical production. In fact, in the presence of O_2_, these transform the hydroxyl radical, the superoxide anion, and other radicals into hydrogen peroxide and organic hydroperoxide, then the hydroperoxide reacts with heavy metals such as iron and copper through Haber–Weiss reactions, favoring a further increase in the synthesis of free radicals [46,47].

The UVA rays trigger reactions that favor the activation of certain enzymes such as porphyrins, NADPH oxidase, and riboflavin, favoring the formation of guanine radicals that act on the DNA, creating damage [24,48]. It should be remembered that the role of heavy metals such as iron, mercury, lead, cadmium, and arsenic in the formation of ROS. The paradigmatic reaction for the formation of ROS by metals is that of Haber–Weiss, but it is not the only one [29,49]. Indeed, in addition to the formation of superoxide anion, hydrogen peroxide, and peroxyl radical, arsenic inhibits the activity of some antioxidant enzymes activated by glutathione [50,51]. Furthermore, the lead increases lipid peroxidation and decreases the activity of glutathione and SOD [52,53,54]. 

In the DNA, the ROS can cause degradation or modification of bases, breaks in both a single strand and both, deletions or translocations of DNA segments, and bonds with other proteins [55,56]. All these factors can determine carcinogenesis, aging, neurodegenerative diseases, autoimmune diseases, and cardiovascular diseases [55,57,58,59,60].

The most important ROS-mediated damage is attributable to the formation of 8-hydro(deoxy)-guanosine (8-OH-G). This is formed by the hydroxyl radical and can modify the expression of some transcription factors by altering gene transcription. Normally, 8-OH-G is expressed in the mitochondria and this would explain the greater insult that the mitochondria receive in the case of oxidative stress [5,50]. Furthermore, 8-OH-G inhibits the activity of TATA boxes, proteins that determine the initiation of DNA transcription [61,62].

The increase in ROS also causes DNA methylation, resulting in the silencing of some genes and altering the ability to repair DNA from damage [11]. ROS act on lipids by promoting their peroxidation. This leads to an alteration of membrane permeability, inactivation of the receptors and enzymes present on it, and alters the phospholipid bilayer [3,11,50]. 

The isoprostanes, substances similar to prostaglandins, are considered markers of lipid peroxidation. At the protein level, ROS causes the breakdown of peptide chains, alteration of the protein charge, cross-links with other proteins, and increases the susceptibility to degradation by proteolytic enzymes [17,63].

### 1.2. Antioxidant Substances

The endogenous antioxidants have the task of counterbalancing the effect of the pro-oxidizing substances and can be divided into the following two categories: enzymatic and non-enzymatic antioxidants [44,64,65]. Among the enzymatic antioxidants, there is SOD, which has three subforms: CuZn-SOD, Mn-SOD, and extracellular SOD (EC-SOD). They are found inside the mitochondria and are all three expressed in the lungs, and their role is to reduce the superoxide anion [15,16]. Other substances with antioxidant activity are catalases, which transform H_2_O_2_ into water and oxygen, glutathione peroxidase (GTPx) that using glutathione converts H_2_O_2_ into two water molecules and detoxifies the peroxidized lipids, and glutathione transferase (GSTs), which inactivates reactive metabolites such as epoxides, aldehydes and hydroperoxides [31,56].

Furthermore, among the enzymatic antioxidants, there are thioredoxins, enzymes containing thiol groups, present at the mitochondrial level and expressed at the pulmonary level, where they perform a protective action against ROS [4,64]. All of these enzymes require NADPH as a reducing species to work [66]. Among the non-enzymatic antioxidants, vitamin E is the most important. Its active form is alpha-tocopherol, which is contained in cereals, dried fruit, cold-pressed oils, and animal fats such as butter and milk [67]. It is the main defense against membrane damage induced by oxidative stress and is able to inhibit the formation of ROS and cause apoptosis of cancer cells [68,69]. The action of vitamin E is closely related to that of vitamin C. This in fact converts vitamin E back to its active form and also has a direct role in reducing ROS [70,71].

Carotenoids, contained in the pigments of vegetable substances, also act as antioxidants [72]. For example, beta carotene reacts with the hydroxyl radical, the superoxide anion, and the peroxyl radical. It also has an anti-inflammatory activity since it inhibits the NF-kB pathway that produces IL-1, IL-6, and TNF-alpha [73,74]. Other polyphenols present in vegetables such as resveratrol and hydroxytyrosol possess powerful antioxidant properties, as shown in animal models and human studies [75,76,77,78,79].

## 2. Pediatrics Syndromes Associated with Oxidative Stress

### 2.1. Fetal Alcohol Spectrum Disorders 

FASD is the term adopted to identify the group of symptoms affecting cognitive and behavioral functioning that can be found in newborns whose mothers consumed alcohol during pregnancy [80,81,82]. It may be interesting to stress that, even if the latest *Diagnostic and Statistical Manual of Mental Disorders* (DSM-V) does not consider FASD to be a specific clinical mental health disorder, it does acknowledge the need for further studies on neurobehavioral disorders associated with prenatal alcohol exposure [83,84].

FASD is a disease with no genetic origin but is due solely and exclusively to the mother’s consumption of alcohol during pregnancy [85]. At present, a minimum dose determining the disease has not yet been identified. For this reason, it is absolutely forbidden to consume alcohol during pregnancy, even in minimal quantities [86].

The effect of alcohol consumption can vary extremely [87], ranging from mild intellectual and/or behavioral impairments to fetal alcohol syndrome (FAS), whose prevention has become the main goal of public health workers since the identification of the damages alcohol consumption can provoke to the fetus [88]. For instance, it should be noted that FAS has been classified as the leading cause of mental retardation worldwide and that, among neurobehavioral and developmental abnormalities, FAS would benefit the most from prevention—that is to say, the foremost preventable cause [84,89,90,91,92,93,94,95].

An interesting aspect of the matter is comparing the causal link between oxidation and FASD as opposed to other genetic disorders. In fact, the patients with FASD are conceived as perfectly healthy, and the oxidation provoked by alcohol is the cause of the pathology since it produces its effect on an otherwise physiological substrate [84,96,97].

More specifically, the scientific community has noticed a strong effect of alcohol exposure on the hippocampal proteome, culminating with the alternation of more than 600 hippocampal proteins playing important roles in the axonal growth regulation, such as annexin A2, nucleobindin-1, and glypican-4, regulators of cellular growth and developmental morphogenesis and, in the cerebellum, cadherin-13, reticulocalbin-2, and ankyrin-2 [98].

The ethanol contained in alcoholic beverages is first converted into acetaldehyde by various enzymes such as alcohol dehydrogenase (ADH), cytochrome CYP2E1, and catalase [99,100]. In the brain, the two most important pathways seem to be those mediated by cytochrome CYP2E1 and catalase, which metabolize 20% and 60% of ethanol, respectively [100]. On the other hand, the most represented pathway for the liver and stomach is mediated by alcohol dehydrogenase, which metabolizes 90% of the ethanol taken [101].

The acetaldehyde produced is then converted into acetate by acetaldehyde dehydrogenase (ALDH), which in turn is converted into acetyl-coenzyme-A in the liver [102].

A major problem of alcohol intake during pregnancy is related to the fact that the fetus has limited or null abilities in the metabolization and elimination of alcohol. Indeed, the various enzymes involved gradually increase their activity during the various stages of gestation [84]. For example, cytochrome CYP2E1 increases its activity compared to an adult, from 40% in the second semester to 80% in the third semester [84]. Therefore, the fetus is likely to be able to metabolize less than half of the alcohol taken by the mother [100].

The increase in ROS in FASD also appears to be due to NOX enzymes belonging to the NADPH-dependent family of enzymes [103]. The NOXs enzymes are expressed at the level of microglia, astrocytes, and the vascular system at the cerebral level, with an important role in the appropriate brain development [104]. The isoforms most involved in ROS production are NOX2 and NOX4 [100]. In FASD patients, it would appear that early exposure to ethanol during pregnancy would increase the activity of NOX isoforms with a significant increase in ROS, cell damage, and ultimately apoptosis [100]. This pathway, in conjunction with that related to the reduced fetal detoxification activity of CYP2E1, would seem to explain the drastic increase in ROS and the consequent phenotype of FASD patients [105].

Even if the mechanisms of the alcohol-induced neuropathology in regions of high vulnerability remain to be comprehensively determined, the teratogenic effects are thought to be the ultimate result of the ethanol-induced dysregulation of a variety of intracellular pathways, which ultimately culminate in toxicity and cell death [85]. The generation of ROS as the possible result of ethanol exposure produces an imbalance in the intracellular redox state, leading to an overall increase in oxidative stress [106]. This would explain the predominant effect that alcohol has on the brain regarding neurobehavioral impairment and deficient brain growth since brain tissue is rich in fatty acids, which chemically are the perfect substrate for the ROS [86]. The most valid biochemical explication would be that the enzyme CYP2E1, whose presence in the brain overlaps with its organogenesis, oxidizes ethanol, generating a hydroxyethyl or superoxide radical, which would target polyunsaturated fatty acid side chains in brain tissue membranes [98,105]. 

As a consequence, fetal brain tissue results in damage during organogenesis, manifesting neurological dysfunctions after birth [86,104,105,107]. 

#### Therapeutic Approach for Oxidative Stress in FASD

Although the best therapeutic strategy is to eliminate alcohol from the diet during pregnancy, several substances have been proposed to reduce the production of ROS in these patients. Among these the astaxanthin, which is a powerful natural antioxidant contained in the carotenoid pigment from a study conducted on mouse models by D. Zheng et al., it was found that the combination of astaxanthin and alcohol during pregnancy did not cause alterations in the growth of the fetus, thus mitigating the harmful effects of alcohol [108]. It also appeared to improve glutathione peroxidase expression and reduce H_2_O_2_ production [108,109]. 

Vitamin E also seems to have a protective role against exposure to ethanol, in fact, it would mitigate the neurotoxicity of ethanol [68,110]. Vitamin C would likewise reduce the production of ROS at the fetal level with a reduction of the disturbances produced by ethanol. Indeed, it would reduce growth retardation and ethanol-mediated neuronal damage [70,108]. At the same time also omega-3 fatty acids thanks to their antioxidant activity would reduce neuronal damage mediated by alcoholic exposure, in fact, what was seen in a mouse model is that fatty acids would increase the production of glutathione with less neurotoxicity from ethanolic exposure [108,111,112].

These data, albeit encouraging, have unfortunately not yet been applied to human models and therefore further studies are needed to evaluate their real efficacy also on humans.

### 2.2. Williams-Beuren Syndrome

Williams-Beuren Syndrome (WBS) is a rare genetic disease with multisystemic involvement, affecting nearly 1 out of 7500–10,000 people. A deletion of a group of genes, situated on chromosome 7 (7q11.23), including the ELN (elastin; OMIM *130,160) gene, is the WBS cause. These genes encode elastin, whose deletion is responsible for the cardiovascular traits and accelerated aging in patients with this disease [113,114]. The common features of this syndrome are represented by the following: facial dysmorphisms, cardiovascular malformations, endocrinological alterations, and intellectual and cognitive disturbances [113]. WBS individuals often show signs of mildly accelerated aging such as cataracts, graying of hair during adolescence, high-frequency sensorineural hearing loss, senile emphysema, premature wrinkling of the skin, and a precipitous age-associated decrease in episodic memory [114,115,116]. The oxidative stress role in these patients is less known than in other genetic disorders; nevertheless, recent studies have shown a correlation between the elastin-insufficiency of WBS and cardiovascular and respiratory diseases [117,118]. For the cardiovascular system, the major impact of ROS seems to be in the hypertension predisposition, in fact as stated before the ELN is a crucial component of the vascular wall providing recoil to elastic vessels [117,118]. 

Arteries with decreased ELN content are less compliant and develop structural modifications that include elevated numbers of smooth muscle and elastic lamellae; consequently, people with ELN deficiency have developmentally, rather than environmentally, elicited vascular stiffness [119]. They also show anatomical differences in branching and arterial tortuosity that lead to a turbulent flow and increased hemodynamic stress on vessel walls [119]. Recently, increasing pieces of evidence demonstrate that ROS production, particularly O_2_^•−^ and H_2_O_2_, through activation of vascular NADPH oxidases, has a central role in vascular mechanotransduction [120,121]. Smooth and endothelial muscle cells express different NADPH oxidases consisting of multiple oxidases and regulatory subunits [122,123]. Other lines of evidence suggest also that hemodynamic forces can either directly or indirectly activate vascular NADPH oxidase-derived ROS production [122,123]. Abundant findings suggest that hypertension might be associated with the potentiated activity of the vascular NADPH oxidases type 1 and 2 having a regulatory subunit, the defined p47phox. This molecule is encoded by the NCF1 (neutrophil cytosolic factor 1) gene located on the telomeric region of chromosome 7 [118]. The lower expression of NCF1 is related to lower blood pressure and minor ROS production, so if the deletion on chromosome 7 extends to include the NCF1 gene, the incidence of hypertension decreases [118,124].

In the respiratory system, WBS subjects with ELN haploinsufficiency may be predisposed to the early development of pulmonary emphysema, the elastin in fact, is a key component of elastic fibers within the lung [124]. Emphysema, a subtype of chronic obstructive pulmonary disease (COPD), is characterized by progressive destruction and loss of elastic fibers, but this is not the only possible mechanism [116]. In WBS, there is a mitochondrial dysfunction. In fact, in the primary fibroblasts of patients affected, decreased basal respiration and maximal respiratory capacity were found, as well as increased ROS generation and decreased ATP synthesis [117]. This mitochondrial dysfunction may be due to the loss of DNAJC30, a gene included in the WBS critical region (WBSCR) [125]. Recent studies have uncovered significant mitochondrial signatures in chronic lung diseases, perturbations of cellular homeostatic programs associated with mitochondrial dysfunction in chronic lung diseases include modulation of the cellular autophagy program, its mitochondria-specific subtype (mitophagy), and associated changes in mitochondrial dynamics and activation of cell death pathways such as apoptosis and necrosis [102,126,127,128]. In addition, mitochondrial dysfunction may have differential and cell type-specific functional consequences in different lung cell types (e.g., epithelial cells, fibroblasts, immune cells), which may differentially impact disease progression, leading to divergent outcomes such as the development of fibrosis or emphysema [129].

Mitochondrial dysfunction could also be related to impaired brain development. Removal of DNAJC30 in mice resulted in hypofunctional mitochondria, diminished morphological features of neocortical pyramidal neurons, and altered behaviors reminiscent of WBS [5]. The mitochondrial features are consistent with our observations of decreased integrity of oxidative phosphorylation supercomplexes and ATP-synthase dimers in WBS [130]. Thus, we identify DNAJC30 as an auxiliary component of ATP-synthase machinery and reveal mitochondrial maladies as underlying certain defects in brain development and function associated with WBS [125].

#### Therapeutic Approach for Oxidative Stress in Williams-Beuren Syndrome

The role of ROS in WBS hypertension pathophysiology could have a therapeutical perspective. The use of generic antioxidants such as vitamins C and E has not shown any positive impact on WBS hypertension [131,132]. However, after recognizing the NOX pathway as a specific etiology, murine studies have been developed. Mutated mice chronically treated with NOX 1 and 2 inhibitors, the apocynin, showed a lowering in systolic blood pressure when compared to the control group [118].

At the same time, the use of the ROS scavenger 2,2,6,6-tetramethyl piperidinyl-1-oxy (TEMPOL) has been used to successfully treat hypertension in two rodent models [31,133]. So specific inhibition of NOX 1 and 2 pathways could lead to pharmacological control of WBS hypertension. 

### 2.3. Ataxia-Telangiectasia

Ataxia-telangiectasia (A-T) is an autosomal recessive disease eliciting several pathologies in the first two decades of life, including immunodeficiency, insulin resistance, telangiectasias, cerebellar ataxia, T-lymphoid tumors, and radiosensitivity [5,134]. Among these, T-cell malignancies and cerebellar ataxia are the most incapacitating phenotypes of this disease [5]. A-T is due to mutations in the *Ataxia Telangiectasia Mutated* (*ATM*) gene [135]. The gene encodes a serine/threonine protein kinase belonging to the phosphoinositide 3-kinase (PI3K)-related protein kinase family [135]. *ATM* plays a main role at the beginning of cellular responses to DNA double-strand breaks [136,137]. Nevertheless, some of the phenotypic disruptions observed in A-T individuals are not easily elucidated only by changes in DNA damage response (DDR) paths [136,137]. Since the antioxidant treatment of *ATM*-null mice improves intrinsic defects in stem cell renewal and might contribute to the delay of their tumor onset, it has been hypothesized that augmented accumulation of intracellular ROS, associated with ATM impairments, may contribute to these diseases [138,139].

The first possible mechanism described is mitochondrial dysfunction. In fact, the ATMs role ATM in maintaining mitochondrial functionality is well shown [102,140,141]. The ATM loss in vivo produces mitochondria disruptions, causing overproduction of ROS, a marked reduction in ATP, and ultrastructural abnormalities [139]. Furthermore, the selective exclusion of impaired mitochondria, known as mitophagy, is strongly impaired, leading to dysfunctional organelles accumulation [5]. The following other pieces of evidence have been reviewed also in neuroblastoma: the depletion of ATM produces comparable mitochondrial phenotypes and mitophagy changes [141]. Studies on thymocytes isolated from mutated mice showed mitochondrial abnormalities in ATM-deficient thymic cells, with disorganized structure and swollen appearance as well as a complex I activity significantly decreased that led to an abnormal generation of ROS taken together, demonstrating the relation between ATM deficiency and mitochondrial dysfunction [5,140].

Noteworthy A-T findings on alternative ROS sources were presented by Weyemi et al., who demonstrated that NADPH oxidase 4 (NOX4) in A-T cells could be a key instrument of oxidative stress [142]. Indeed, it was shown that NOX4, which constitutively triggers ROS in a variety of tissues and cell types as well as has a subtle role in oxidative DNA damage and the consequent senescence, was quite up-regulated in A-T individuals, comparable to normal cells with an ATM protein kinase privation [28,143]. Additionally, NOX4 is correlated with higher oxidative damage and apoptosis [6,142]. Moreover, NOX4 inactivation decreased cancer incidence (lymphoma) in ATM-deficient mice compared to control mice [6,142].

Semlitsch et al. demonstrated the following other important sources of damage in A-T patients: a specific oxidation product: the ox-LDL (oxidized low-density lipoprotein) [144,145]. OxLDL is a potent proinflammatory chemoattractant for macrophages and T-lymphocytes. It is also cytotoxic for endothelial cells and stimulates them to release soluble inflammatory molecules [146]. In addition, oxLDL has turned out to be highly immunogenic and promotes changes in cell cycle protein expression and subsequent translocation and activation of transcription factors [145,147]. These events help to perpetuate a cycle of vascular inflammation and lipid/protein dysregulation within the artery wall and also may create a cellular prothrombotic state that complicates later stages of atherosclerosis [147].

OxLDL generates oxidative stress in the vascular system induced phosphorylation of ATM and downstream activation of p21 in fibroblasts and endothelial cells. ATM-deficient cells are extremely sensitive to the toxic effects of ROS, especially H_2_O_2_ and nitric oxide developing an increased DNA fragmentation [148,149].

In conclusion, oxidative stress in A-T is related to various sources such as mitochondrial dysfunction, augmented production of ROS by the up-regulation of NOX4, and increased sensibility to ox-LDL with DNA fragmentation [6,145]. 

#### Therapeutic Approach for Oxidative Stress in Ataxia-Telangiectasia

Due to the important oxidative stress in A-T, different antioxidant agents have been tested in A-T mice. One of these agents is TEMPOL (4-hydroxy-2,2,6,6-tetramethylpiperidin-1-oxyl) [137,150]. When administered continuously in the mice diet, TEMPOL resulted in an increased lifespan by prolonging the latency of thymic lymphomas. In addition, TEMPOL treatment reduced ROS, restored mitochondrial membrane potential, and reduced tissue oxidative damage in vitro-cultured A-T cells.

Another promising molecule is N-acetyl cysteine (NAC) [150,151]. A-T mice have increased levels of 8-OH deoxyguanosine, an indicator of oxidative DNA damage. When fed with NAC supplementation, these mice showed reduced values of 8-OH deoxyguanosine, similar to the level in wild-type mice. NAC has been demonstrated to be able to prolong the lifespan and reduce both the incidence and multiplicity of lymphomas. Other antioxidants have been studied, such as tetramethylisoindolin-2-yl oxyl (CTMIO) and salen-manganese complexes, but with a less clear correlation with disease modification [150].

### 2.4. Down Syndrome

Down syndrome (DS) is a congenital disorder caused by a complete or partial trisomy of Chr21 (HSA21), and it is the most common genetic cause of significant intellectual disability, with an incidence of around 1:800 births [152]. Most DS cases (95%) are caused by non-disjunction of chromosomes in meiosis I during the formation of gametes, while 3.2% are caused by translocation, and 1.8% of residual DS cases are caused by mosaicism [153]. The effects of trisomy 21 can be very different from one individual to the next, and not every DS subject shows the same phenotypic features. The main alteration is represented by intellectual disability, other common features are congenital heart disease, Alzheimer’s disease, leukemia, hypotonia, motor disorders, and various physical anomalies [154]. Although pathological mechanisms leading to DS phenotypes are still uncertain, it is evident that the presence of the third chromosome 21 is responsible for altered development during embryogenesis and organogenesis [155]. Many of its clinical features were also studied as possible consequences of oxidative stress and cellular senescence since the change in chromosome 21 affects genes playing key roles in the redox state regulation [117,156]. When associated with other redox imbalance genetic diseases, DS has been broadly investigated [157,158]. Several findings showed that changes in proteins and genes involved in ATP consumption, mitochondrial pathways, and increased ROS production may explain the wide variety of phenotypes [159]. 

The most important genes that are involved in the increase in oxidative stress levels found in DS individuals and in the Ts65Dn mouse model are *SOD1*, *APP*, *BACH1*, *Et2*, *S100B*, and *CRB* [160]. 

The first chromosome 21 gene that was characterized and identified in different DS tissues was SOD-1 (Cu/Zn superoxide dismutase 1) (OMIM *147,450) [161]. It acts as an antioxidant defense that catalyzes the dismutation of superoxide radicals (O_2_^−^) to hydrogen peroxide (H_2_O_2_), then metabolized to water by catalases (CAT) and glutathione peroxidase (GTPx) [162]. SOD-1 was shown to be approximately 50% higher than normal in a wide range of DS tissues and cells, including B and T lymphocytes, fibroblasts, and erythrocytes [163]. Furthermore, SOD-1 overexpression in the brain was not associated with a parallel CAT and GTPx elevation, determining an imbalance in the ratio of SOD-1 to CAT and GTPx levels, leading to an H_2_O_2_ accumulation and consequent damage [164]. Surprisingly, DS tissues, including the brain, show changes in the SOD-1/GTPx activity ratio [165]. Associated with CAT and GTPx, a decreased expression of peroxiredoxin 2 was also found in the DS fetal brain, which contributes to the improved susceptibility of DS neurons to undergoing oxidative damage [166]. Since SOD-1 may play an important role in the pathogenesis of Ts21, it could be used as a potential biomarker for the prenatal diagnosis of Ts21 [167]. 

In addition to the well-recognized role of SOD-1, alterations in the oxidative imbalance could also be caused by the over-production of beta-amyloid (Aβ), due to the triplication of APP. APP (Amyloid Beta A4 Precursor Protein) (OMIM *104,760) encodes for a precursor protein of Aβ. Its overexpression leads to Aβ deposition and increases the formation of senile plaques, a main neuropathological finding of Alzheimer’s disease [154]. DS is the most common cause of early-onset Alzheimer’s disease-dementia [168]. In patients with DS, the increase in APP expression is strongly associated with Aβ deposition in adult life and the early and increased formation of senile plaques [158]. Oxidative stress and early plaque formation in the brain are closely connected. In fact, ROS damage increases the probability of the formation of protein aggregates as it obstructs the normal processes of protein elimination. The Aβ aggregates can be targets of oxidative processes, inserted as oligomers within the cell membrane and promoting a process of lipid peroxidation (LPO) [117]. 

BACH1 (BTB domain and CNC homolog 1; OMIM *602,751), encoded on Hsa21, is another key element in the regulation of the antioxidant response in DS [159]. It is a transcription repressor that inhibits selected gene transcription involved in stress response, such as heme oxygenase-1 (HO-1) and NADPH. The BACH1 overexpression potentiates ROS production from the endothelial cell’s mitochondria [159]. DS mouse model studies and investigations into DS patients demonstrated that BACH1 was significantly upregulated [169]. In DS, it is probable that BACH1 protein upregulation could block the induction of antioxidant genes, therefore eliciting increased oxidative stress in the cell [163]. 

The analyses of two more genes, S100β and Ets-2, both located on chromosome 21, have been reported as associated with oxidative damage. 

S100β (S100 calcium-binding protein, beta; OMIM *176,990) is an astroglia-derived Ca2+-binding protein actively secreted from astrocytes that modulates the activity of neurons, microglia, astrocytes, monocytes, and endothelial cells depending on its concentration [170]. S100β increased expression in astrocytes from DS and Alzheimer’s patients was shown in association with neuritic plaques [171]. The overexpression of S100β increases ROS formation and results in increasing neuritic neuronal and APP with consequently accelerated amyloid accumulation [117]. 

Ets-2 (ETS Protooncogene 2, Transcription Factor) (OMIM *164,740) is a transcription factor playing crucial roles in immune responses, cancer, and bone development, and it is overexpressed in Down Syndrome [155]. Overexpression is associated with increased neuronal apoptosis [117]. Ets-2 overexpression in cultured HCN leads to activation of a mitochondrial death apoptotic pathway. In DS/AD brains, upregulation of ets-2 appears closely associated with AD neurodegenerative lesions. Chronic oxidative stress in DS and AD brains may promote ets-2 expression, which may predispose to the activation of a mitochondrial death pathway [172]. 

In conclusion, cognitive and neurological disorders may be the consequence of the overexpression of SOD1, APP, ETS-2, S100β, and abnormal production of BACH1. Finally, the following other main alterations that can affect individuals with DS are congenital and acquired cardiovascular abnormalities: although no specific gene has been identified, it seems that increased oxidative stress and mitochondrial dysfunction are associated with the increased development of these complications [173]. 

#### Down Syndrome Therapy

Since oxidative stress in individuals with DS has been associated with trisomy of the 21st chromosome resulting in DS phenotype as well as various morphological abnormalities, different studies have been performed to study the impact of antioxidant interventions [156]. 

The effects of antioxidants on oxidative brain damage have been investigated in several studies on a canine animal model of aging. In 2002, Cotman et al. found that after 1–6 months of administration of D,L-α-tocopherol, carnitine, D,L-α-lipoic acid, ascorbic acid, and other dietary antioxidants, the ability of spatial attention was improved [174].

Some oligo-elements such as zinc and selenium can decrease oxidative stress biomarkers and decrease inflammatory cytokines [175]. The study of the effects of these antioxidant elements has obtained controversial results. In 1989, Lockitch et al. demonstrated that zinc (25–59 mg/day) administered for 6 months had no effect on lymphocyte functions but eased daily cough [176]. Selenium (10 µg/kg/day) administered for 6 months increased levels of IgG and decreased infections, and a dose of 25 µg/kg/day administered from 0.3 to 1.5 years increased activity of GPx and reduced the SOD/GPx ratio [177]. During the last decades, different natural antioxidants have been studied. 

Lott et al. (2011) daily administered αtocopherol (900 IU), ascorbic acid (200 mg), and α-lipoic acid (600 mg) to 53 individuals with DS over 40 years of age and Alzheimer’s disease for two years, and they showed that the antioxidant therapy with vitamin E and C was effective, safe, and tolerable [178]. Nevertheless, the neuropsychological result assessments showed no difference in the cognitive improvement of these individuals [179]. In 2008, Ellis et al. obtained similar results in a study with 156 DS children who were supplemented with antioxidants, including a reduced form of folic acid [180]. On the other hand, Lockrow et al. evaluated the supplementation of vitamin E in a DS model of oxidative stress in the rat brain. The antioxidant supplementation not only decreased oxidative stress markers in the brain but also improved spatial memory performance and attenuated cholinergic neuropathology. They concluded that vitamin E delays the onset of cognitive and morphological abnormalities in DS animal models and that this may represent a safe and effective treatment early in the progression of neuropathology associated with [181]. Another study from Shichiri et al. demonstrated the effectiveness of vitamin E in an animal model of DS. They have demonstrated that chronic therapy with a-tocopherol in pregnant mothers was able to decrease lipid peroxidation products and improve cognition in newborns [182]. In 2014, Parisotto et al. proposed that antioxidant therapy with vitamins C and E, which lacks adverse effects at the doses here used, might bring clinical benefits to DS children that may somehow improve cognitive function according to the effect found in a murine model, or at least attenuate neurodegenerative conditions in those patients [181]. So, despite the evidence of the presence of oxidative stress in individuals with DS, it is unclear the reason for failure in antioxidant interventions. The reasons could be an inappropriate choice of antioxidants, and inadequate dose, or duration of administration. 

As for new DS therapy, since it has been demonstrated that physical activity reinforces antioxidant resistances and decreases lipid peroxidation in adults and aged people without DS, many studies have tried to understand if there could be some benefits for people affected by DS [183]. Indeed, physical activity induces increased antioxidant defense in younger as well as older subjects [156,184]. Furthermore, stronger physical exercise potentiates antioxidant events in the heart, liver, and skeletal muscles [185]. Unfortunately, the effects in DS people are not crystal since their physiological responses to exercise diverge from the general population [186].

Shields et al. in 2017, conducted a systematic review to research the effect of regular exercise on oxidative stress in DS individuals. In particular, six databases were searched, and studies were included if DS participants (any age) had completed an exercise program of at least 6 weeks duration and at least one biomarker measured the generation or removal of reactive oxidative species. The main finding of the review was that no clear pattern emerged for how individual biomarkers of oxidative stress or antioxidant activity responded to regular short-term aerobic exercise programs. In fact, if there is strong evidence that regular exercise decreases biomarkers of oxidative stress and increases antioxidant activity in people in the general population, there is no similar evidence for people affected by DS, and randomized controlled trials did not propose recommendations regarding the type and dose of exercise that could be beneficial to people with this condition [187]. 

In summary, oxidative stress implications in the DS phenotype have been demonstrated, although the direct cause-and-effect association between the increase in oxidant-related damage and DS clinical outcomes has not yet been clearly established [160]. The use of controlled supplementation with antioxidants, physical activity, and physical exercise could improve cognition and comprehensively benefit people with DS.

### 2.5. Marfan Syndrome 

Marfan syndrome (MFS) is an autosomal dominant disease that affects the connective tissue with variable penetrance and an estimated prevalence of one in 10,000–20,000 individuals [188]. In 90% of cases [189], MFS is caused by mutations in Fibrillin-1 (FBN1), located on chromosome 15q21.1 and containing 65 exons [190]. The cardinal features involve the ocular and skeletal systems (e.g., tall stature, arachnodactyly, and ectopia lentis) [191], but the most life-threatening manifestations are related to cardiovascular complications, including mitral valve prolapse, arrhythmias, coronary artery disease, left ventricular hypertrophy, congestive heart failure, aortic insufficiency, dilatation of the aortic root, and aortic dissection [192,193]. Considering those cardiovascular complications, early recognition and appropriate management are critical for patients with MFS. Clinical criteria and, in particular, Ghent nosology, outlined in 2010, are used in the diagnosis of MFS [194].

It is well known that the highly reactive oxygen-derived free radicals (ROS) play an important role in the genesis and progression of various cardiovascular diseases, including arrhythmias, aortic dilatation, aortic dissection, left ventricular hypertrophy, coronary arterial disease, and congestive heart failure [195]. MFS is characterized by the presence of ascending aortic aneurysms resulting from the altered assembly of extracellular matrix fibrillin-containing microfibrils and dysfunction of TGF-β signaling [196]. It has been demonstrated that patients affected by MFS show impaired contractile function and endothelial-dependent relaxation resulting from oxidative stress in the thoracic aorta [197]. Endothelial dysfunction increases the inducible nitric oxide synthase (iNOS) pathway, leading to an excess in nitric oxide (NO) production that causes tissue damage [198]. Moreover, the results of studies have suggested that ROS could be involved in smooth muscle cell phenotype switching and apoptosis as well as matrix metalloproteinase activation, resulting in extracellular matrix (ECM) remodeling [199]. Among ROS species, Jiménez-Altayó et al. identified that H_2_O_2_ directly produced by NOX4 and/or by the transmutation of O_2_^•−^ by SODs could be the most relevant ROS candidate for the Marfan-associated redox stress because unlike O_2_^•−^, H_2_O_2_ is permeable to cell membranes and has a significantly longer lifespan than O_2_^•−^, showing high reactivity for cysteine residues leading to their oxidation [200]. Oxidative stress plays an important role in the formation of the ascending aortic aneurysm but also in the evolution of the aneurysm. Studies on both animals and humans studies demonstrated that augmented redox stress is correlated to the progression of the aortic aneurysm [199]. According to some studies, the progression of thoracic aortic aneurysm is the result of the markedly impaired aortic contractile function as well as decreased nitric oxide (NO)-mediated endothelial-dependent relaxation [201]. Recently, Fiorillo et al. demonstrated, for the first time, signs of oxidative stress in the plasma of patients with MFS. Moreover, they showed a significant correlation between the intensity of oxidative stress and the severity of the clinical manifestations, suggesting systemic oxidative damage [202]. 

#### Therapeutic Approach for Oxidative Stress in MFS

The identification of crucial aspects responsible for aneurysm formation could be used for the development of advanced therapies to decrease the susceptibility of aneurysm formation or to slow aneurysm growth [203,204]. Resveratrol is a potent polyphenol that can diminish oxidative stress [205]. It is capable of decreasing vascular senescence via the inhibition of nicotinamide adenine dinucleotide phosphate (NADPH) oxidase activity and consequently decreasing oxidative stress in a SIRT1-dependent fashion [206].

*Hibiscus sabdariffa* Linne (HSL) is commonly used against hypertension, pyrexia, inflammation, liver disorders, and kidney and urinary bladder stones. It also has an antibacterial, antifungal, hypocholesterolemic, antispasmodic, and cardioprotective effect [207,208]. Many studies have described how the organic acids from *Hibiscus sabdariffa* Linne calyces increase cellular antioxidant capacity and decrease oxidative stress [209]. Soto et al. demonstrated that infusion of HSL allows an increase in antioxidant capacity of both the enzymatic and nonenzymatic systems in the plasma of MSF patients [210].

*Allium sativum* (garlic) has antioxidant properties too [211]. Perez-Torres examined if garlic could affect the OS of patients with MFS. They administered 500 mg of cursive sativum Chinese garlic tablets (Deodorized Garlic) 2 times per day for 2 months in MFS patients and pointed out that treatment with garlic could lower the OS threshold by increasing the activity of antioxidant enzymes and could help in the prevention and mitigation of adverse OS in patients with MFS [212].

In conclusion, since oxidative stress plays an important role in the pathogenesis of the cardiovascular life-threatening manifestation of MFS, the aim of the treatment of patients with MFS should involve obstructing the progression of aortic dilation with the use of agents with antioxidant properties. The application of antioxidants could help in the prevention and reduction of oxidative stress in MFS patients, with a positive effect on patient survival.

### 2.6. Fanconi’s Anemia 

Fanconi’s anemia (FA) is an inherited pathology of DNA repair, with progressive pancytopenia, bone marrow failure, variable congenital malformations, and a predisposition to hematological or solid tumors [213]. AF is transmitted with an autosomal recessive inheritance and is due to mutations in genes involved in DNA repair and genomic stability. In total, 15 genes have been identified, these 15 genes encode proteins identified as FANCA, B, C, D1, D2, E, F, G, I, J, L, M, N, O, and P. These proteins form nuclear complexes and are activated in response to DNA damage breakage [214,215,216].

The incidence is approximately 1–5 cases per 1,000,000 inhabitants, with approximately 2000 cases described worldwide [213]. The phenotypic anomalies are many, such as aplasia of the radius, skin hyperpigmentation, microphthalmia, nystagmus, and reduced vision. Cardiac, renal, and urogenital defects and short stature, deafness, and hypogonadism can occur in a lower percentage of incidence [213]. The prognosis is unfavorable due to the higher incidence of solid tumors and leukemia [217].

In patients with FA, oxidative stress is an important determinant of some clinical conditions related to the disease. In fact, it has been seen that in affected patients there would be a reduction in the activity of SOD with consequently reduced production of antioxidant species. Furthermore, there would seem to be an increase in the production of TNF-α, which would lead to an increase in the production of O_2_^•−^ with consequent apoptosis and disruption of DNA [218,219]. In fact, the increase in TNF-α would seem to lead to reduced activity of some proteins of the FANC group such as FANCA and FANCG with a reduced ability to prevent DNA damage [220]. The latter protein is expressed at the mitochondrial level and its mutation would lead to the reduced activation of a mitochondrial peroxide, PRDX3, with important antioxidant activities [219]. The FANCC protein, on the other hand, would bind cytochrome-P450 2E1 (CYP2E1), which has a detoxifying activity at the cellular level [220]. The FANCD2 protein interacts with the ATM protein and appears to stabilize it by reducing the production of radical species [218].

#### Therapeutic Approach for Oxidative Stress in Fanconi’s Anemia

From a therapeutic point of view, attempts have been made to intervene in the increase in ROS production in various ways. For example, through the intake of quercetin in the diet, which belongs to the flavonoid family, normally present in fruit and vegetables, there would be a reduction in the production of radical species [215]. Another therapeutic strategy is to intervene by administering inhibitors of double-stranded RNA-dependent Kinase (PKB), which would lead to a reduction in the activation of the inflammatory response [221]. In another study conducted by Cappelli E et al., the reduction of ROS production was evaluated using a combination of quercetin, C75, and rapamycin with a reduction of lipid peroxidation [222]. However, further studies are needed to evaluate therapeutic opportunities.

### 2.7. Autism Spectrum Disorders (ASD)

Autism spectrum disorders are defined as a group of neurodevelopmental disorders that involve alterations in social, work, school, and personal functioning [223]. They usually cause difficulties in acquiring, maintaining, and applying specific skills or sets of information [224]. The current prevalence in Italy is 15 children per 1000 with a 1:4 male-to-female ratio. In the last decade, there has been an increase in diagnoses due in part to the change in diagnostic criteria [225]. The etiopathogenesis is unknown and it is therefore believed that it may be multifactorial and dependent on both environmental and genetic factors. In recent years, the correlation between environmental pollutants and autism spectrum disorders has been studied, and what has emerged is that exposure to such substances as ionizing radiation, pesticides, and heavy metals increases the risk of developing this pathology [226]. From a clinical point of view, the symptoms are variable but usually characterized by the following: persistent deficits in social communication and interaction, repetitiveness, and sectorial behavior, interests, or activities [224]. The diagnosis is clinical and is based on the evaluation of the patient and on the execution of some tests that are generally carried out from the age of two, including the Autism Diagnostic Interview-Revised (ADI-R); Childhood Autism Rating Scale (CARS) and Autism Diagnostic Observation Schedule (ADOS) [225]. Regarding oxidative stress, patients with ASD seem to have higher levels than the general population [227]. This would seem to be mainly due to a reduction in the activity of endogenous antioxidants such as SOD, GTPx, and CAT [228]. The reduction of the enzymatic activity would seem to increase the production of pro-oxidizing and lipid peroxidation-related substances such as F2-isoprostane and 8-iso-prostaglandin F2α [229]. Furthermore, there would seem to be an alteration in metallothionein-3 (MT-3), a protein with detoxifying activity expressed in the brain. This, in fact, seems to be reduced in patients with ASD, causing greater neuronal toxicity [227]. It should also be mentioned that, in general, in patients with ASD, there is a reduction in total glutathione with consequently reduced detoxification of pro-oxidizing metabolites [229].

#### Therapeutic Approach for Oxidative Stress in ASD

As far as the therapeutic approach is concerned, there are currently no data in the literature on humans, but only on animal models. Among the antioxidants proposed as therapeutics are the following: sulforaphane, coenzyme Q10, N-Acetyl-Cysteine, omega-3 fatty acids, arachidonic acid, and resveratrol [230,231,232,233]. All of these substances, with the exception of resveratrol, have a beneficial role in reducing oxidative stress and the most effective would appear to be N-Acetyl-Cysteine [231]. Some studies conducted on children with ASD have evaluated how a diet rich in foods known to be antioxidants, such as citrus fruits, leafy vegetables, and dark chocolate, has a benefit on the reduction of pro-oxidizing substances at the plasmatic level [226]. Although the data seems encouraging, further studies are needed to confirm them.

### 2.8. Primitive Immunodeficiencies 

Primitive immunodeficiencies (PID) represent a wide group of diseases, generally considered as conditions with an increased rate of serious and recurring infections caused by an alteration in the immune response. In the immuno-dysregulated scenario, along with the augmented infection susceptibility, there is also a manifestation of autoimmunity. As said, under the definition of PID, there are numerous clinical syndromes [234]. However, the correlation between these syndromes and oxidative stress has been sufficiently studied only in a few of them, as follows: the common variable immunodeficiency disease (CVID), the most serious subtype of severe combined immunodeficiency (SCID), the reticular dysgenesis (RD), and the chronic granulomatous disease (CGD) [235].

#### 2.8.1. Common Variable Immunodeficiency

CVID is the most common symptomatic PID in adults [235]. It is characterized by a variable clinical spectrum. The most common clinical manifestations are represented by hypogammaglobulinemia, recurrent infections, autoimmune diseases, lymphoid malignancy, enteropathy, and granulomatous diseases [235]. The CVID symptoms usually onset in childhood, reaching their full form in the second or third decade of life [235]. 

In the last few years, the role of oxygen metabolism as a possible actor in the pathophysiology and clinical manifestations of CVID has been studied [236]. Basaranoglu et al. in one of the latest human studies showed a reduced level of serum catalase (CAT) in a patient with symptomatic CVID when compared with the control group [236]. In addition, the CAT levels were lower in CVID patients with recurrent infections and autoimmune disorders than in CVID patients with only recurrent infections. These data could suggest some abnormalities in responses to ROS [236]. Other evidence of increased oxidative stress in CVID comes from the study of Aukrust et al., where a significantly higher level of reduced homocysteine was found in CVID patients compared to the control group [237]. A recent study showed how selenium median levels and glutathione peroxidase (GPX) activity were significantly lower in CVID patients compared to controls [238]. Furthermore, there was a higher percentage of high values of C-reactive protein, higher concentrations of oxidized LDL, and lower concentrations of Apolipoprotein A in the group of CVID patients compared to the control group [238]. All these data suggest increased oxidative stress and cardiovascular risk in these patients [238,239]. 

#### 2.8.2. Therapeutic Approach for Oxidative Stress in CVID

No studies are available for the therapeutical use of antioxidants in this disease. However, due to the importance of Selenium as an antioxidant defense mediated by the glutathione peroxidase (GPX) family protein against dyslipidemia and cardiovascular disease, and due to the Se reduced level in CVDI patients, a diet integration could be useful [238]. Nevertheless, further studies are needed. 

#### 2.8.3. Severe Combined Immunodeficiency (Reticular Dysgenesia)

Severe combined immunodeficiency (SCID) is a group of rare disorders caused by mutations in different genes involved in the development and function of infection-fighting immune cells [240]. In this group of diseases, peripherical T-cells are absent and, as a consequence, serious or life-threatening infections, especially viral infections, which may result in pneumonia and chronic diarrhea, frequently occur [241]. Moreover, mycotic infections such as *candidiasis* of the mouth and airways or *Pneumocystis jirovecii* pneumonia are common [240]. The possible phenotypes of these disorders are SCID with only T-cells lacking (but with a normal number of B-cells) or a deficiency of both B and T- cells. Furthermore, phenotype can present an absence of natural killer (NK) cells [241]. RD is the most severe form of human SCID. It is associated with mutations in adenylate kinase 2 (AK2), the mitochondrial intermembrane space enzyme that regulates energy metabolism [242]. Rissone et al. studied how AK2 deficiency in the zebrafish model affected hematopoietic stem and progenitor cell development with increased oxidative stress and apoptosis [242]. AK2-deficient arrests the maturation of the myeloid line at the promyelocyte stage with an increased AMP/ADP ratio, indicative of an energy-depleted adenine nucleotide profile [242]. The amount of oxidative stress produced in the mitochondria is mitigated in normal cells by a defense system that scavenges and detoxifies from the ROS [242]. Mitochondrial dysfunction can lead to excessive ROS production that saturates the cellular antioxidant capacity [242,243]. In embryonal RD cells, it was found to have a significantly increased level of superoxide, especially in mitochondria [240]. This is similar to the abnormalities that can be observed in human fibroblasts derived from RD patients [240]. These results demonstrate that AK2 deficiency induces oxidative stress in hematopoietic tissues, with augmented apoptosis in all cell lines, acting as a pathogenic substrate [242].

#### 2.8.4. Therapeutic Approach for Oxidative Stress in RD

The antioxidant-only treatment in animal models of RD is unlikely to have strong long-term beneficial effects on RD [242]. However, it can mitigate the increased myeloid and lymphoid line apoptosis, raising the number of lymphocyte and neutrophil cells, as demonstrated in zebrafish models [240]. However, no human studies are yet available. 

#### 2.8.5. Chronic Granulomatous Disease

The CGD is a rare genetic disorder (1/200,000 live births) caused by a mutation in genes encoding the NADPH-oxidase, which compromises the ability of phagocytes to generate ROS [244]. This leads to higher susceptibility to fungi and bacteria infections that are resistant to non-oxidative killing methods of immune cells [245,246]. Along with the infection susceptibility, CGD patients also frequently suffer from severe and debilitating inflammatory conditions [246]. This aspect of the disease is correlated with enhanced basal sterile inflammatory signaling and pro-inflammatory cytokine production by the CGD immune cells [245]. Even lacking NADPH-oxidase, recent studies have shown a paradoxical increased oxidative status in CGD phagocytes [247]. These cells display clear signs of oxidative stress, including an induced expression of antioxidants and an altered oxidation of cell surface thiols [244]. The outcome of this augmented oxidative stress has been found in mitochondrial dysfunction [244]. The ROS pathological increase explains why CGD patients have a chronic sterile inflammation, one of the ununderstood aspects of this disease. In fact, mitochondria-derived ROS (mtROS) enhanced the phosphorylation of ERK1/2 and induced the production of IL8 and other pro-inflammatory cytokines in myeloid cells [244]. The increased oxidative stress has been found also in human cells of CGD patients showing high content in oxidated lipids, proteins, DNA (8-hydroxy-2′-deoxyguanosine (8-OHdG)), and other oxidation products [247]. In addition, significantly decreased coenzyme Q10 plasmatic concentration has been observed in CGD patients [247]. Coenzyme Q10 is a lipophilic molecule ubiquitously present in cell membranes and particularly abundant in the mitochondrial electron chain [248]. Coenzyme Q10 acts as a direct antioxidant of the cell membrane but is also able to prevent oxidative tissue damage indirectly by regenerating other antioxidants, such as ascorbic acid or alpha-tocopherol. Coenzyme Q10 deficiency in mitochondria leads to respiratory chain dysfunctions such as the ones found in CGD [245,249]. Furthermore, coenzyme Q10 modulates the function of the immune cell response [247]. Different studies demonstrate how a coenzyme Q10 deficiency can be closely related to an increase in inflammation resulting from mitochondrial ROS overproduction, as seen in fibromyalgia patients [248,250]. Analogous mitochondrial pathologies have been noticed in CGD cells, so it can be hypothesized that coenzyme Q10 deficiency could play a role in the pathogenesis of this primary immunodeficiency disorder [245]. 

#### 2.8.6. Therapeutic Approach for Oxidative Stress in CGD

At present data indicate that unfavorable oxidant/antioxidant balance is a feature of CGD and causes a predisposition to developing a hyperinflammatory state in innate immune cells [249]. Considering low coenzyme Q10 content and oxidative stress in the plasma of patients with CGD, a possible therapeutical strategy could be the supplementation of coenzyme Q10 in this group of patients [245].

### 2.9. Gaucher Disease

Gaucher disease (GD) is one of the most common lysosomal storage diseases and it is caused by mutations in the gene GBA1. GBA1 encodes the enzyme glucocerebrosidase (GCase, acid-β-glucosidase) that leads to altered glucocerebrosidase activity, resulting in the accumulation and storage of glycosphingolipids as the glucocerebroside (GL1) in the lysosomes of macrophages. GL1-engorged macrophages (Gaucher cells) are usually found in the bone marrow, spleen, liver, lungs, lymph nodes, and other organs of patients with GD [251,252].

Glucosylsphingosine (GlcSph, lyso-GL1, lyso-Gb1) is another glycosphingolipid that is present in much lower concentration but probably more pathogenic than GL1 itself in patients with GD [253]. There are the following three different subtypes of GD, according to the presence of neurological deterioration, age at identification, and disease progression rate: the most common is represented by GD1 (non-neuronopathic); GD2 (acute neuronopathic), and GD3 (chronic neuronopathic), which are less common but are associated with a more severe phenotype [254]. GD is normally suspected for the presence of unexpected anemia, thrombocytopenia, and organomegaly. The gold standard for the diagnosis of GD is the presence of low enzymatic activity of GBA1 in peripheral blood compared with normal controls [255,256].

There are the following four treatments available for GD1: 3 ERTs and 1 SRT. No drugs have been approved for GD2 or GD3 [251]. Despite the current therapies, chronic pain and fatigue are two of the most important symptoms associated with GD1. This is probably due to the chronic inflammation associated with the release of pro-inflammatory cytokines and other mediators either directly from Gaucher macrophages or indirectly by crosstalk between Gaucher cells and immunomodulatory lymphocytes [257]. However, it has been reported that the accumulation of toxic glycosphingolipids within cells leads to the production of reactive oxygen species (ROS) and an imbalance between the pro-oxidants and the antioxidant reserve, resulting in oxidative stress and inflammation [258].

The loss of the function of GCase is in fact responsible for an important decrease in the mitochondrial membrane potential, adenosine diphosphate phosphorylation, and an increase in oxidative stress and fragmentation of mitochondria [259]. For this reason, different authors have tried to study the relation between oxidative stress and GD. Rollo et al. investigated the relationship between ROS and GD by analyzing blood oxidative stress markers in GD patients submitted to ERT at different stages during the treatment. They discovered that RT-treated GD patients showed an improvement in antioxidant capacity, which was further increased just after recombinant enzyme infusion [260]. Mello et al. showed an alteration in the concentration of CAT, SOD, and SH, which suggests that there was a change in reactive oxygen species in GD type I patients when compared to HC. This increase in CAT, SOD, and sulfhydryl could likely be related to the prevention of the increase in hydrogen peroxide, preventing damage to lipids [258].

Recently, Kartha et al. decided to study the levels of multiple oxidative stress biomarkers in plasma and red blood cells from untreated patients, in stable individuals undergoing standard-of-care therapy, and in healthy controls.. They found significant differences in key oxidative stress biomarkers in untreated patients compared to healthy controls, while in treated patients, results generally fell between the controls and the untreated patients [255].

#### Therapeutic Approach for Oxidative Stress in GD

At present time, data indicate that patients with GD have higher levels of biomarkers of OS. Therefore, those findings could be considered for evaluating the use of antioxidants as adjunctive therapies for patients with GD1.

## 3. Conclusions

As it emerged from the scientific literature, oxidative stress is correlated to an alteration between antioxidant species and oxidant species, with an increase in the latter.

As regards the genetic syndromes studied, the role of oxidative stress is certainly important in their clinical and pathological manifestations, as shown in Table 2. The interesting thing to note is that, for example, in the case of FASD, an alteration in the oxidizing species due to alcohol consumption is the factor that causes the disease itself. On the contrary, in other syndromes such as Williams syndrome, Down syndrome, Marfan syndrome, Gaucher syndrome, ataxia-telangiectasia, autistic spectrum disorder, Fanconi’s anemia, and primitive immunodeficiencies, the oxidative stress is secondary to the genetic alterations that cause the syndromes.

This is crucial in terms of clinical relevance as it can help us understand how the elimination of oxidizing factors during pregnancy, such as alcohol, can totally prevent FASD. At the same time, in the other aforementioned syndromes, where the damage from oxidative stress is secondary to the underlying pathology, the use of supplements of substances with antioxidant activity and strengthening the importance of a balanced diet and practicing sports could counteract the damage induced by oxidative stress.

However, further studies and new scientific evidence are needed to fully understand the mechanism related to oxidative stress in the context of genetic pathologies in order to think about acting therapeutically in the future.

## Figures and Tables

**Figure 1 antioxidants-11-01983-f001:**
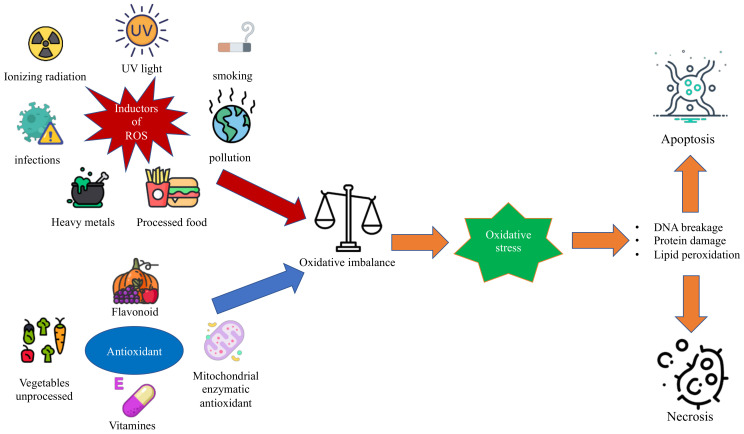
The figure shows the oxidative imbalance between ROS and antioxidant factors that leads to oxidative stress and the related consequences on the cell.

**Table 1 antioxidants-11-01983-t001:** The table shows the reactions implicated in the production of oxidative species.

ION/RADICAL: [4,20,21,22]	REACTION: [4,23,24,25]	ENZYME/COFACTORS: [26,27,28]
SUPEROXIDE ION	O_2_ + e^−^ → O_2_ ^•−^	
HYDROGEN PEROXIDE	2O_2_^•−^ + 2H^+^ ⇔ O_2_ + H_2_O_2_	SUPEROXIDE DISMUTASE
HYDROXYL RADICAL	H_2_O_2_ + e^−^ → HO^−^ + ^•^OH	FENTON REACTION
SINGLET OXYGEN	2H^+^ + 3O_2_ → ^1^O_2_ + H_2_O_2_	NADPH
NITRIC OXIDE	2 L-arginine + 3 NADPH + 3 H^+^ + 4 O_2_ → 2 L-citrulline + 2 NO + 3 NADP^+^ + 4 H_2_O_2_	NITRIC OXIDE SYNTHASE
PEROXYNITRITE RADICAL	NO + O_2_^•−^ → NO(O_2_)^−^	

**Table 2 antioxidants-11-01983-t002:** The table shows the different clinical features and pathways leading to an increase in oxidative stress in pediatric syndromes.

Pediatric Syndromes:	Genetic Mutations:	Clinical Features:	Alterations of Genes and Proteins Involved in Oxidative Imbalance:	Clinical Manifestations:
Ataxia-telangiectasia	Loss of function of ATM gene on chromosome 11q22.3	Telangiectasia, Cerebellar ataxia, Immunodeficiency, Insulin resistance, Radiosensitivity, T-lymphoid tumors	PI3K (Phosphoinositide 3-kinase) [139]: a protein involved in cell growth and survival, his loss causes alterations in DNA damage response NOX4 (NADPH oxidase 4) [142]: an enzyme involved in the production of ROS is higher express OxLDL (oxidized low-density lipoprotein) [146]: is a proinflammatory chemoattractant for macrophages and T-lymphocytes	Telangiectasia cerebellar ataxia Telangiectasia, accelerated aging, promoter of T-lymphoid tumors Prothrombotic alterations and atherosclerosis
Autism spectrum disorders (ASD)	The responsible genes have not been found	Deficits in social communication and interaction, Repetitiveness and sectorial in behavior, interests or activities	MT-3 (Metallothioneine-3) [228]: a protein with detoxifying activity expressed in the brain is reduced in patients with ASD determining neuronal toxicity SOD-1 (Cu/Zn superoxide dismutase 1) [162]: is a protein that acts as an antioxidant defense and transforms the superoxide radicals into H_2_O_2_, it is reduced in patients with ASD GTPx (Glutathione peroxidase) [261]): is an enzyme that uses glutathione to convert H_2_O_2_ into two water molecules and detoxifies the peroxidized lipids and glutathione transferase (GSTs), which inactivates reactive metabolites such as epoxides, aldehydes, and hydroperoxides, it is reduced in patients with ASD	Accelerated neuronal death Neurological alterations, cell death Neurological alterations Cell death
Chronic granulomatous disease (CGD)	Mutations of the genes encoding for NADPH-oxidase	Recurrent infections, Recurrent abscess in the liver, gastrointestinal tract, lymph nodes and lungs, HypergammaglobulinemiaAnemia	NADPH-oxidase [18]: this enzyme is reduced in these patients determining a reduction in the production of ROS from phagocytes Coenzyme Q10 [248]: this coenzyme produces antioxidant substances, it is reduced in patients with CGD	Recurrent bacterial and mycotic infections Recurrent bacterial and mycotic infections
Common variable immunodeficiency disease (CVID)	Unknown mutations in 90% of cases. In 10% of cases are related to *TNFRSF13B* gene	Hypogammaglobulinemia, Recurrent infections, Autoimmune diseases, Lymphoid malignancy, Enteropathy granulomatous disease	CAT (catalase [17]): this enzyme catalyzes the decomposition of H_2_O_2_ to H_2_O and O_2_, its level is reduced in patients with CVID determining an overproduction of ROS GTPx (Glutathione peroxidase) [261]: this enzyme uses glutathione to convert H_2_O_2_ into two water molecules and detoxifies the peroxidized lipids and glutathione transferase (GSTs), which inactivates reactive metabolites such as epoxides, aldehydes, and hydroperoxides, it is reduced in patients with CVID	Promote carcinogenesis Recurrent infections, promote carcinogenesis
Down syndrome (DS)	Trisomy of chromosome 21	Intellectual disability, Facial dysmorphism, Congenital heart disease. Anticipated Alzheimer disease. Leukemia, Hypotonia, Neurodevelopmental disorders	SOD-1 (Cu/Zn superoxide dismutase 1) [162]: this proteins acts as an antioxidant defense and transforms the superoxide radicals into H_2_O_2_, in DS is higher than normal, but this augmentation is not accompanied by the increases in catalases (CAT). and glutathione peroxidase (GTPx) with an accumulation of H_2_O_2_ APP (Amyloid Beta A4 Precursor Protein) [155]: is a protein with an over-production in these patients that leads to an increase in beta-amyloid (Aβ) BACH1 (BTB domain and CNC homolog 1) [169]: inhibits genes involved in cell stress response. In DS is upregulated determining the production of ROS. S100β (S100 calcium-binding protein) [171]: is a Ca2+ binding protein produced from astrocytes modulating the activity of endothelial cells microglia, and neurons. It is overexpressed in DS. Ets-2 (ETS Protooncogene 2, Transcription Factor) [172]: is a transcription factor involved in bone growth, immune response, and cancer. It is overexpressed in DS with increased neuronal apoptosis.	Accelerated aging, neurological alterations Early onset of Alzheimer’s disease Accelerated aging, neuronal alterations, intellectual disability Accelerated amyloid deposition, early onset of Alzheimer’s disease Alzheimer’s disease, neurological alterations and disability
Fanconi’s anemia (FA):	Mutations in genes involved in DNA repair and genomic stability, 15 genes have been identified	Aplasia of the radius, Skin hyperpigmentation, Microphthalmia Nystagmus, Reduced vision, Cardiac, renal and urogenital defects short, Stature deafness, Hypogonadism	SOD-1 (Cu/Zn superoxide dismutase 1) [162]: is a protein that acts as an antioxidant defense and transforms the superoxide radicals into H_2_O_2_, it is reduced in patients with FA TNF-α (Tumor necrosis factor alpha) [219]: is a cytokine involved in systemic inflammation and is a member of a group of cytokines that stimulate the acute phase reaction. It is increased in patients with FA and this would lead to an increase in the production of O_2_^•−^ PRDX3 (Peroxiredoxin 3) [220]: is an enzyme localized in mitochondria with antioxidant function, it is reduced in patients with FA	Accelerated cell deathApoptosis and disruption of DNA, promote carcinogenesisOverproduction of ROS determining DNA damage
Fetal alcohol spectrum disorders (FASD)	Related to alcohol exposure of fetus during pregnancy	Facial anomalies, Growth deficiency, Neurobehavioral impairment, Deficient brain growth, Non-febrile seizures	NOX2 and NOX4 (NADPH-dependent enzymes) [118,143]: are a family of proteins that produce ROS they are expressed in microglia, neurons, astrocytes, and in brain vessels. They are overexpressed in FASD. CYP2E1 (Cytochrome P450 isoform 2E1) [104]: is an enzyme involved in the metabolism of ethanol, it is reduced in the fetus and this determines the overproduction of ROS	Neurological anomalies Neurological anomalies
Gaucher disease	Mutations in GBA1	Anemia, Fatigue, Neurological deterioration, Organomegaly	ROS [262]: accumulation of toxic glycosphingolipids cells leads to the production of reactive oxygen species	Asthenia Fatigue
Marfan syndrome	Mutations in the gene for fibrillin-1 (*FBN1*)	Skeletal abnormalities (dolichostenomelia, arachnodactyly, scoliosis, chest wall deformity, tall stature, ligamentous laxity, abnormal joint mobility, and protrusio acetabulae, scoliosis), Ectopia lentis Mitral valve disease, Dilatation of the aortic root, Aortic dissection	H_2_0_2_ [32]: H_2_O_2_ directly produced by NOX4 and/or by the transmutation from O_2_^•^^−^ by superoxide dismutase (SODs) Endothelial dysfunction [188]: it increases the inducible nitric oxide synthase (iNOS) pathway leading to an excess in nitric oxide (NO)	Aortic aneurysm Aortic aneurysm
Reticular dysgenesis		Viral infections such as pneumonia, Mycotic infections such as candidiasis, Chronic diarrhea	AK2 (Adenylate kinase 2) [240]: it is a mitochondrial intermembrane space enzyme that regulates energy metabolism, it is reduced in patients with RD leading to an overproduction of ROS.	Maturation impeding of myeloid cells at the promyeloid stage Accelerated apoptosis
Williams–Beuren syndrome	Microdeletion 7q11.23	Facial dysmorphism, Aortic stenosis, Neurodevelopmental delays, Accelerated aging, Cocktail party personality	ELN (Elastin) [119]: encodes for elastin an important protein detectable in the vascular wall, which provides recoil to elastic vessels. It is present also in the elastic fibers of the lungs. DNAJC30 (DnaJ Heat Shock Protein Family Hsp40 Member C30) [125]: encodes for a mitochondrial protein involved in the removal of damaged complex I subunits. His loss determines an alteration in ATP synthase resulting in increased cell apoptosis	Cardiovascular alterations, pulmonary emphysema and accelerated aging Chronic lung disease, alterations in neocortical pyramidal neurons and altered behaviors

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
