# Peer review of "The Impact of Oxidative Stress on Pediatrics Syndromes"

_antioxidants, 2022, doi:10.3390/antiox11101983_

Round 1

Reviewer 1 Report

I would like to compliment the authors of this review for engaging in such an interesting and relevant topic. However I do have several concerns with the current version of this manuscript. I am aware that this is not a systematic review of the literature, but more an expert type of review. That said, I find that the overall balance of the article (selection of syndromes, manner of presentation) it a times a bit biased.

In the introduction the authors explain the concept of oxidative stress and reactive oxygen species at the biochemical, physiological and pathophysiological level. But, after reading this first part of the manuscript, I still did not get a comprehensive view on the topic. Some of the paragraphs read very “haphazardly”. Most of the information is there, yet it is not always structured in a very logical or relevant manner. I sometimes had the impression that it was written by a biochemist with a focus on the basic science, rather that the clinical implications.

In table 1, I think there are mistakes related to the hydroxyperoxide radical, the chemical formula that is written down is identical to the one for the hydroxyl radical; where I think it should be a different one resulting in °HO2.

My second issue is with the selection of the syndromes that have been discussed. To my knowledge there is also evidence of impact of the oxidative stress pathways in relation to autism spectrum disorder, common variable immunodeficiency syndrome, Marfan syndrome and many others. It is unclear to me how the author selected the specific syndromes that they describe in this manuscript. For a comprehensive review I would expect at least one paragraph that mentions systematically all of the other syndromes where this pathological pathway is implicated (in a table for instance).

The third issue is with the part on the fetal alcohol spectrum disorder. There are almost 10 paragraphs in this part of the manuscript but only two of them really discuss the scientific findings related to oxidative stress and the fetal metabolism and the pathological effects. Most of the other paragraphs are a very pronounced expression of the dangers of alcohol consumption during pregnancy. As a paediatrician I do find this message extremely important, but as a scientist I expect a review (on antioxidants) to be more objective. If the author's would like to highlight the dangers of alcohol consumption in pregnancy in so many paragraphs, that message would be better suited in a review article about the dangers of prenatal alcohol consumption. The affiliation of one of the authors is solely to two societies that focus on alcohol consumption during pregnancy and fetal effects (SITAC and SIFASD). Again these are very important issues with major societal impact, but their message in a review on oxidative stress should be limited.

Some of the pathophysiological effects of alcohol that are mentioned in this part are finding of experiments in mouse or rat models that only study adults specimens. There is no caveat in the text that states that these findings may be different in fetal organisms or that the data from animal studies cannot always be fully translated to humans. In all the references related to this part of the manuscript I find very few that are specific for the fetal context. The final paragraph regarding possible interventions and clinical use of antioxidants during pregnancy are interesting yet the authors seem to have given up on prevention of this syndrome which in my opinion is too ‘fatalistic’.

The other syndromes are much better described. However again there is a lot of text devoted to the therapy regarding Down syndrome while this is not is present for the Williams Burlington syndrome and ataxia teleangiectasia syndromes.

Author Response

Answers to the comments raised by reviewer 1

I would like to compliment the authors of this review for engaging in such an interesting and relevant topic.

Reply: we do thank the reviewer for the positive comments.

However, I do have several concerns with the current version of this manuscript. I am aware that this is not a systematic review of the literature, but more an expert type of review. That said, I find that the overall balance of the article (selection of syndromes, manner of presentation) it a times a bit biased.

Reply: yes, we presented a paper as an expert type of review. According to the criticism of the reviewer, we updated the paper to avoid biases in the selection of syndromes and manner of presentation by including further syndromes and shortening the FASD section.

In the introduction the authors explain the concept of oxidative stress and reactive oxygen species at the biochemical, physiological and pathophysiological level. But, after reading this first part of the manuscript, I still did not get a comprehensive view on the topic. Some of the paragraphs read very “haphazardly”. Most of the information is there, yet it is not always structured in a very logical or relevant manner. I sometimes had the impression that it was written by a biochemist with a focus on the basic science, rather that the clinical implications.

Reply: we would like to stress the point that the journal Antioxidants provides an advanced forum for studies related to the science and technology of antioxidants focusing on new insights and ideas on active species and processes of biological relevance, natural products, mechanisms of action, applications and uses. It is not a specific clinical journal as the brother MDPI journal “Children”. However, according to the comment of the reviewer we revised and shortened this specific section.

In table 1, I think there are mistakes related to the hydroxyperoxide radical, the chemical formula that is written down is identical to the one for the hydroxyl radical; where I think it should be a different one resulting in °HO2.

Reply: we do apologize for the typing mistake leading to a repetition. Accordingly, the reaction was deleted.

My second issue is with the selection of the syndromes that have been discussed. To my knowledge there is also evidence of impact of the oxidative stress pathways in relation to autism spectrum disorder, common variable immunodeficiency syndrome, Marfan syndrome and many others. It is unclear to me how the author selected the specific syndromes that they describe in this manuscript. For a comprehensive review I would expect at least one paragraph that mentions systematically all of the other syndromes where this pathological pathway is implicated (in a table for instance).

Reply: we do thank the reviewer for her/his suggestion. Accordingly, other pediatric syndromes were discussed and Table 3 (now Table 2) was updated and improved (pages 38-41 of the revised paper).

The third issue is with the part on the fetal alcohol spectrum disorder. There are almost 10 paragraphs in this part of the manuscript but only two of them really discuss the scientific findings related to oxidative stress and the fetal metabolism and the pathological effects. Most of the other paragraphs are a very pronounced expression of the dangers of alcohol consumption during pregnancy. As a paediatrician I do find this message extremely important, but as a scientist I expect a review (on antioxidants) to be more objective. If the author's would like to highlight the dangers of alcohol consumption in pregnancy in so many paragraphs, that message would be better suited in a review article about the dangers of prenatal alcohol consumption. The affiliation of one of the authors is solely to two societies that focus on alcohol consumption during pregnancy and fetal effects (SITAC and SIFASD). Again these are very important issues with major societal impact, but their message in a review on oxidative stress should be limited.

Reply: yes, it is true, many of us published several papers of FASD (preclinical and clinical studies) and the extended written text is based on this wide knowledge. However, as suggested we shortened the FASD section to avoid imbalances with the description of the other syndromes. 

Some of the pathophysiological effects of alcohol that are mentioned in this part are finding of experiments in mouse or rat models that only study adults specimens. There is no caveat in the text that states that these findings may be different in fetal organisms or that the data from animal studies cannot always be fully translated to humans. In all the references related to this part of the manuscript I find very few that are specific for the fetal context. The final paragraph regarding possible interventions and clinical use of antioxidants during pregnancy are interesting yet the authors seem to have given up on prevention of this syndrome which in my opinion is too ‘fatalistic’.

Reply: as stated before the FASD was shortened and revised.

The other syndromes are much better described. However again there is a lot of text devoted to the therapy regarding Down syndrome while this is not is present for the Williams Burlington syndrome and ataxia teleangiectasia syndromes.

Reply: as suggested, we included in the revised version of the paper additional information on the therapy regarding Williams-Beuren syndrome and A-T syndromes. We also included further information about the Marfan syndrome, Gaucher syndrome, Autistic Spectrum Disorder, Fanconi’s anemia and Primitive Immunodeficiencies (text highlighted in light yellow in the revised paper).

Reviewer 2 Report

The authors of “The Impact of Oxidative Stress on Pediatrics Syndromes” have done a good and interesting bibliographic work on the correlation of some pediatric syndromes with oxidative stress. I would have to make only a few minor comments.

- First of all, why have the authors focused on these syndromes and not on others?

- In line 124 “MPO” should be defined, although later it is in the list of abbreviations.

- The example on line 137 could be in the same previous paragraph since it is the same idea.

- As in the previous case, define the acronyms FAS, FASD and PFAS in the title of table 2.

- The use of “one” ("one should", normally) should be changed. Although it is understood, it is strange in reading.

- The references in table 3 should go to each gene or protein, not in the header of each column.

- Correct the H2O2 subscripts in table 3.

Author Response

Answers to the comments raised by reviewer 2

The authors of “The Impact of Oxidative Stress on Pediatrics Syndromes” have done a good and interesting bibliographic work on the correlation of some pediatric syndromes with oxidative stress. I would have to make only a few minor comments.

Reply: we do thank the reviewer for the positive comments

- First of all, why have the authors focused on these syndromes and not on others?

Reply: according to the comments of both reviewers we included additional information on other pediatric syndromes characterized by oxidative stress to avoid biases in the selection of syndromes and manner of presentation.

- In line 124 “MPO” should be defined, although later it is in the list of abbreviations.

Reply: as requested we defined MPO (page 4, lines 108 and 125 of the revised manuscript, text highlighted in yellow).

- The example on line 137 could be in the same previous paragraph since it is the same idea.

Reply: as suggested, the sentence was revised (page 4, lines 132-140 of the revised manuscript, text highlighted in yellow).

- As in the previous case, define the acronyms FAS, FASD and PFAS in the title of table 2.

Reply: according to this comment and to the comments of the other reviewer the chapter on FASD was revised and shortened and Table 2 was deleted (sections 2.1. and 2.1.1 of the revised manuscript, text highlighted in yellow).

- The use of “one” ("one should", normally) should be changed. Although it is understood, it is strange in reading.

Reply: we apologize for the abuse of the term “one”. Accordingly, we revised the text.

- The references in table 3 should go to each gene or protein, not in the header of each column.

Reply: as requested, Table 3 (now Table 2, pages 38-41) was updated and improved

- Correct the H2O2 subscripts in table 3.

Reply: we corrected the text by using subscripts

Round 2

Reviewer 1 Report

The authors have delivered an impressive effort to extend the paper and include several other relevant pediatric syndromes. They have addressed all relevant reviewer comments.